# Transcriptional Regulation of Natural Killer Cell Development and Functions

**DOI:** 10.3390/cancers12061591

**Published:** 2020-06-16

**Authors:** Dandan Wang, Subramaniam Malarkannan

**Affiliations:** 1Laboratory of Molecular Immunology and Immunotherapy, Blood Research Institute, Versiti, Milwaukee, WI 53226, USA; dwang2@Versiti.org; 2Department of Microbiology and Immunology, Medical College of Wisconsin, Milwaukee, WI 53226, USA; 3Department of Medicine, Medical College of Wisconsin, Milwaukee, WI 53226, USA; 4Department of Pediatrics, Medical College of Wisconsin, Milwaukee, WI 53226, USA

**Keywords:** NK cell, development, transcription factors, IL-2, IL-15, IL-21, IL-12

## Abstract

Natural killer (NK) cells are the major lymphocyte subset of the innate immune system. Their ability to mediate anti-tumor cytotoxicity and produce cytokines is well-established. However, the molecular mechanisms associated with the development of human or murine NK cells are not fully understood. Knowledge is being gained about the environmental cues, the receptors that sense the cues, signaling pathways, and the transcriptional programs responsible for the development of NK cells. Specifically, a complex network of transcription factors (TFs) following microenvironmental stimuli coordinate the development and maturation of NK cells. Multiple TFs are involved in the development of NK cells in a stage-specific manner. In this review, we summarize the recent advances in the understandings of TFs involved in the regulation of NK cell development, maturation, and effector function, in the aspects of their mechanisms, potential targets, and functions.

## 1. Introduction

Natural killer (NK) cells are cytotoxic lymphocytes that mediate anti-viral and anti-tumor responses [1,2,3]. Unlike T or B cells, NK cells exert their cytotoxic functions without prior sensitization utilizing germline-encoded, non-rearranged activating receptors [1,2]. NK cells primarily mediate their functions through four distinct effector mechanisms. NK cells recognize target cells for killing via four mechanisms: natural cytotoxicity, antibody-dependent cell cytotoxicity (ADCC) [4], TNF-related apoptosis-inducing ligand (TRAIL) [5], and Fas ligand (FasL) [6]. NK cell effector functions are mediated by the releasing of lytic granules containing perforin and granzymes that cause apoptosis, binding TRAIL or FasL on target cells that also causes apoptosis, and secreting inflammatory and inhibitory cytokines (interferon-γ/tumor necrosis factor-α (IFN-γ/TNF-α) and interleukin (IL)-10 respectively) [7,8]. NK cell functions, including defense against intracellular pathogens, tumor surveillance, and immunoregulation, signify their high clinical relevance. The clinical applications include antibodies (such as Daratumumab, targeting CD38, and Elotuzumab, targeting signaling lymphocytic activation molecule F7 (SLAMF7)) for ADCC [9,10]; genetically-engineered NK cells to enhance their functions against tumors [11,12,13]; and infusion of purified and activated NK cells into patients [14,15,16]. The essential step to realizing these clinical applications is to reach the full functional potential of NK cells in patients or ex vivo.

The commitment, development, maturation, and functions of NK cells rely on a complex matrix of transcription factors (TFs, intrinsic signals) primarily coordinated by environmental cues, including cytokines (extrinsic signals) [17,18]. The role of various cytokines, such as IL-2, IL-12, IL-15, and IL-18, in NK cell biology, has been well-established. The identities of molecules involved in multiple signaling cascades downstream of major activating receptors have been largely defined [19]. However, the identities and the mechanisms by which the transcriptional network operates in NK cells are far from fully understood. This knowledge gap is a formidable challenge that limits the successful clinical utilization of NK cells. The next significant challenge is to fully define the transcriptional networks that regulate the development, heterogeneity, and effector functions of human NK cells. This is essential in order to formulate effective NK cell-based cellular therapies and to augment their effector functions.

Historically, cell surface markers were used to define distinct developmental stages of NK cells. While these approaches led to significant advancements in our understanding of NK cells, they failed to provide information on the associated molecular mechanisms. Furthermore, these methods did not offer any insight into the heterogeneity of developing or mature NK cells. For their part, genetically modified mice only provide insights into the functions of select genes of interest and their downstream targets. A pioneering technique, flow cytometry combined mass-spectrometry, cytometry by time of flight (CyTOF), can detect up to 40 cellular markers simultaneously [20]. Studies using CyTOF demonstrated that there are 6000 to 30,000 distinct NK cell phenotypes within an individual based on 35 different cell surface antigens [21]. However, this approach does not offer any insights into the transcriptional mechanisms that govern the generation of vast heterogeneity among NK cells. In recent years, single-cell RNA-sequencing (scRNA-seq), one of the technological breakthroughs, has allowed scientists to define NK cell heterogeneity and developmental stages based on their transcriptomic profiles at a single-cell level [22,23,24]. Further, sophisticated bioinformatics analyses such as SCENIC (single-cell regulatory network inference and clustering) can be used to map transcriptional regulatory networks based on scRNA-seq data and simultaneously optimizing cellular state identifications using the inferred networks [25]. To define the transcriptional networks, it is essential to recapitulate the functions of known TFs in NK cells. Here, we summarize the TFs that control NK cell development and functions.

## 2. NK Cell Development

Both human and murine NK cells primarily arise from self-renewing pluripotent hematopoietic stem cells (HSCs), which reside in bone marrow (BM) and commit to a sequential order of intermediate progenitors [26,27,28]. Murine NK cells primarily develop in the BM [28]. However, in humans, apart from the BM, other organs such as lymph nodes (LN), spleen, and tonsils also support NK cell development [29,30]. Lymphoid-primed multipotential progenitors (LMPPs), the first precursor arising from HSCs, commit to common lymphoid progenitors (CLPs) (Figure 1A) [31,32]. CLPs give rise to all lymphocyte populations, including Pro-B, Pre-T, innate lymphoid cells (ILCs), lymphoid tissue inducers (LTi), and CD122^+^ Pre-T/early NK cell progenitors (NKPs) [33]. Conventionally, different cell surface markers are used to define distinct developmental stages of NK cells.

In mice, BM-derived HSCs (Lin^−^stem cell antigen (Sca)^+^ CD117 (c-Kit)^+^) [34] differentiate to CLPs defined by Lin^−^Sca^low^CD117^low^CD135 (FLT3)^+^ (Figure 1B) [35]. The expression of CD127 (IL-7Rα), CD27 (Figure 1C), and CD244 indicate the commitment to pre-NKP [36]. The refined-NKPs (rNKP) are defined by the expression of CD122 (IL-2Rβ) [28]. The expression of NKG2D marks a successful commitment to the immature NK cell (iNK) stage [37]. Through the immature stage, NK cells express NK1.1, CD159A (NKG2A), CD159C (NKG2C), CD226 (DNAM-1), and NCR1, along with the cell adhesion molecules CD62L (L-selectin) and CD43 (Leukosialin) [38]. The expression of CD51 (Integrin αV) and CD49b (DX5) marks the early maturation stage of NK cells [38]. Terminal mature NK (mNK cells) are defined as CD43^+^ along with the downregulation of CD27, upregulation of CD11b (Mac-1) and acquiring killer cell lectin-like receptor G1 (KLRG1), from NK1.1^+^CD49b^+^CD43^+^CD27^+^CD11b^−^KLRG1^−^ (early maturation) to NK1.1^+^CD49b^+^CD43^+^ CD27^−^CD11b^+^ KLRG1^+^ (terminally mature) [38,39,40]. KLRG1 is an inhibitory receptor and NK cell terminal maturation marker [41]. The acquisition of distinct Ly49 receptors defines the transition to the mNK stage, where they are functionally licensed [38,42]. Among Ly49 receptors, there are two groups, activating Ly49s (Ly49D and Ly49H) and inhibitory Ly49s (Ly49A, Ly49C, Ly49I, and Ly49G) [43,44,45]. Murine NK cell development occurs in a four-stage process following NK1.1 expression, CD27^−^CD11b^−^ CD27^+^CD11b^−^ → CD27^+^CD11b^+^ → CD27^−^CD11b^+^ [39,40] (Figure 2).

In humans, lineage negative (Lin^−^) CD34^+^ HSCs differentiate into CD45RA^+^CD133^+^ LMPPs [33]. CD244 (2B4) is expressed from pre-NK cell precursors (pre-NKPs) [46], and continues its sustained expression throughout the life-span of NK cells. Along with CD135 (Flk-2) [47], NKPs are characterized by the expression of CD7, CD127 (IL-7Rα), CD122 (IL-2Rβ), CD117 (c-Kit), and IL-1R1^low^ [47]. The immature NK cells (iNK) are defined by higher expression of IL-1R1 along with the expression of CD314 (NKG2D), CD335 (NKp46), CD337 (NKp30), and CD161 (NK1.1). The next is the transitional stage, with the maximal expression of NKG2D, CD335, CD337, and CD161, along with the high expression of CD56 (CD56^bright^). The expression of NKp80 separates this transitional stage to two sub-stages, NKG2D^+^CD337^+^CD161^+^NKG2A^+^CD56^bright^ NKP80^−^ (sub-stage a, which is relatively immature) and NKG2D^+^CD337^+^CD161^+^NKG2A^+^CD56^bright^NKp80^+^ (sub-stage b, which is relatively mature) [48,49,50]. The decreased expression of CD56 and increased expression of CD16, along with the expression of distinct CD158 (KIR) subtypes, marks the NKp80^+^CD56^dim^CD16^+^KIR^−/+^ mature NK (mNK) stage [51] (Figure 3). Human NK cells account for 5–20% of circulating lymphocytes in the peripheral blood mononuclear cells (PBMCs) of healthy adults [52].

## 3. Role of Cytokines in NK Cell Development

The external signals (cytokines) which promote NK cell expansion and functional maturation are well characterized. The expression of cytokine receptors such as CD117 (c-Kit), CD127 (IL-7Rα), and CD122 (IL-2Rβ) defines the developmental stages of NK cells. Cytokine receptors such as common gamma chain (γ_c_) and IL-2Rβ expressed during their developmental stages provide essential signals for NK cell development, homeostasis, and functions [53]. The critical components from these cytokine signaling pathways can act as either upstream regulators or downstream targets of TFs in the transcriptional regulatory network to control NK cell development [54].

### 3.1. SCF, Flt3, c-Kit, and IL-7 Control the Early Commitment of HSCs into LMPPs and CLPs

Cytokines, FMS-like tyrosine kinase 3 ligand (Flt3L, also known as Flk2) and stem cell factor (SCF) promote the commitment of HSCs to CLPs [55,56]. Flt3 and c-Kit are tyrosine kinase receptors and interact with Flt3L and SCF, respectively [57]. Flt3- or c-Kit-deficient mice display a reduction of CLP numbers [55,56,58]. Flt3 deficiency impairs cell proliferation and cell cycle status of HSCs and CLPs [58]. In addition to the defect of HSCs and CLPs, *Flt3L^−/−^* mice display a significant reduction of splenic NK cells, but they are still detectable [56]. Similarly, *c-Kit^−/−^* mice generate fewer NK cells that are poorly cytolytic [59]. Flt3L or SCF alone is unable to drive NK cell expansion and differentiation from CD34^+^ HSCs [60], whereas Flt3L or SCF can synergize with IL-15 to significantly augment NK cell proliferation [60,61]. Furthermore, Flt3L can also induce substantial higher expression of CD122 to enhance the effect of IL-15 signaling [60]. SCF is capable of enhancing MAPK-mediated human NK cell proliferation and functions as an additive to IL-15 [61]. These observations suggest that c-Kit may not be essential for NK lineage commitment but does play a role in NK cell development.

IL-7 is one of the γ_c_ receptors utilizing cytokines [62], and its receptor (IL-7R) is comprised of unique IL-7Rα (CD127) and the γ_c_ subunit (CD132) (Figure 1C). The expression of CD127 marks the end of the CLP stage and the start of the NKP stage [35,36,47]. Irrespective of these observations, *Il7*- or *Il7ra*-deficient mice display severe defects mainly in thymic NK cell development, as well as T and B cell differentiation [63,64]. The peripheral and splenic NK cells from these knockout mice do not exhibit any apparent defects [65]. In humans, IL-7Rα is specifically highly expressed in CD56^bright^ cells and requires interaction with IL-7 to inhibit the apoptosis of the CD56^bright^ subset [66].

### 3.2. IL-15 is An Obligatory Cytokine for the Development and Maturation of NK Cells

IL-15 is required and essential for both the development and survival of NK cells, but not for NK cell lineage specification or commitment [67,68]. The IL-15R complex is composed of three subunits, unique high-affinity alpha chain (IL-15Rα), shared beta chain (IL-2Rβ), and the γ_c_ [69,70,71]. IL-2Rβ is shared between IL-2R and IL-15R complexes [71]. γ_c_ is the shared subunit by IL-2R, IL-4R, IL-7R, IL-9R, IL-15R, and IL-21R [62].

NK cell populations in *Il15^−/−^*, *Il15ra^−/−^*, *Il2rb^−/−^* or γ_c_*^−/−^* mice are severely reduced or absent [70,72]. Overexpression of IL-15 in mice results in upregulated NK cell numbers [73]. These observations suggest that IL-15 and its receptors play an essential role in NK cell maturation and expansion. Intracellular IL-15 binds IL-15Rα to form the complex, which is shuttled to the surface of the trans-presenting dendritic cells (DCs) to NK cells expressing IL-15Rα/IL-2β/γ_c_ heterotrimers [74]. The trans-presenting cells include DCs, macrophages, stromal, and epithelial cells [75]. This unique trans-presentation mechanism explains the reason that conventional NK cells are unable to survive in the BM of *Il15ra^−/−^* mice [74,76,77,78,79]. IL-15 induces the differentiation of human CD34^+^ HSCs into CD3ε^−^CD56^+^ NK cells in vitro [60]. In mice, the IL-15R-mediated signaling pathway is important to direct NKPs into mature NK cells [67], but not required for the generation of NKPs [68]. The few remaining NK cells from IL-15-deficient mice show measurable but reduced cytotoxicity and IFN-γ production in response to YAC-1 target cells and IL-12 stimulation, respectively [68]. For the critical role of IL-15, its downstream signaling molecules STAT5 and JAK3 are also indispensable components in NK cell development [80,81,82]. Similar to IL-15- or IL-15R-deficient mice, development of NK cells in STAT5-deficient mice is blocked after the NKP stage and they are unable to clear tumor cells [81,82].

### 3.3. IL-2 is Essential for NK Cell Proliferation

IL-2, a growth factor for NK cells, acts through either the high-affinity trimeric receptor comprised of IL-2Rα, IL-2Rβ chain, and γ_c_ or intermediate affinity dimeric receptors formed by IL-2Rβ and γ_c_ [83,84]. It is a critical cytokine for NK cell survival, activation, and expansion [85,86,87]. NK cells in IL-2-deficient mice have impaired cytotoxicity and IFN-γ production [85]. IL-2 drives NK cell proliferation and promotes the production of perforin and Granzyme B [86]. This is consistent with the fact that ex vivo NK cell culture requires exogenous IL-2 to activate and systemic IL-2 administration to make them proliferate in vivo and augment their cytotoxicity and cytokine production in patients [88]. However, studies show that the expression of CD11b and Ly49 receptors (mature NK markers) in IL-2-, IL-4-, or IL-7-deficient NK1.1^+^ NK cell populations is comparable to that of wildtype (WT) mice [68]. The IL-2-deficient mice have similar NK cell numbers of different developmental stages and normal capability to produce IFN-γ and kill target cells [68]. These observations suggest that IL-2 is dispensable for both the development and effector functions of NK cells.

### 3.4. IL-21 Synergizes with IL-15 and IL-2 to Augment NK Cell Cytotoxicity

IL-21, acting through IL-21Rα and γ_c_, is employed to expand and stimulate ex vivo human NK cells in the presence of IL-2 and IL-15 in clinical protocols [89,90,91,92]. IL-21 is mainly produced by T helper cells and NKT cells [93], which builds the obligatory link between NK and T cells. IL-21 promotes human NK cell survival in vitro to a similar extent with IL-2 [94]. IL-21 synergizes with IL-2 to augment NK cell cytotoxicity by upregulating the expression of NKp46, NKG2A, perforin, and Granzyme B [94]. In addition, IL-21 synergizes with IL-15 to promote progenitor cells from human BM to expand and enhance NK cells effector functions [95]. Although IL-21 enhances cytotoxicity and IFN-γ production of activated murine NK cells [96], it dampens IL-15-mediated expansion of resting murine NK cells [97], suggesting murine and human NK cells have different responses to this cytokine.

### 3.5. IL-12 and IL-18: Essential Interphase between Myeloid and NK Cells

IL-12 and IL-18 augment NK cell cytotoxicity and enhance IFN-γ production [98]. Both of these cytokines are produced by pathogen-activated DCs and macrophages [99,100]. The crosstalk between the myeloid and NK cells builds an immune response network against tumor or infection [101]. IL-12 is a potent stimulator of IFN-γ production [102,103], which is supported by the fact that the IFN-γ production is significantly reduced in IL-12 knockout mice or anti-IL-12 antibodies [104]. IL-12 functions in a synergetic way with IL-2, IL-15, or IL-18 and considerably enhances the production of IFN-γ in NK cells [98]. In addition to the enhanced cytolytic activity of NK cells against cancer cell lines, IL-12 can also promote their proliferation [105,106]. IL-18, defined as an IFN-γ inducing factor, is one IL-1 cytokine family member [107,108]. NK cells from IL-18- or IL-18R-deficient mice display reduced IFN-γ production and impaired cytotoxicity [109,110], but IL-18 alone is not sufficient to induce IFN-γ production in vitro [109,111]. IL-18 synergizes with IL-2, IL-15, or IL-21 to upregulate IFN-γ production and augment other human NK cell effector functions [112].

## 4. Transcriptional Regulation of NK Cell Development

### 4.1. Notch1 Regulates the Generation of HSCs and Its Commitment to CLP

A number of TFs have been described in both mouse (Figure 4) and human (Figure 5) models that are involved in the development and functions (Figure 6 and Figure 7) of NK cells. The highly conserved Notch family is important in determining cell fate [113,114]. The mammalian Notch family proteins, including Notch1 through Notch4, interact with five ligands (Jagged-1, Jagged-2 and Delta-1, -3, and -4) [115,116,117,118,119], and act as transcriptional activators [120,121,122]. Notch1 and Notch2 are expressed in hematopoietic precursor cells [123,124]. Notably, Notch1 is essential for generating HSCs in the yolk sac where embryonic hematopoiesis begins [125]. After prenatal hematopoiesis occurs in the fetal liver, the Notch signaling pathway is essential to promote T cell commitment in the thymus and suppress alternative lineage fate potentials, including B cells, DCs, and myeloid cells [126,127,128]. However, some studies suggest that Notch signaling does not restrict NK cell lineage specification [129,130,131]. In the absence of lineage specifying factors, such as Pax5 in B cells or Bcl11 in T cells, pro-B cells or thymocytes can be reprogrammed into NK cells when cultured with Notch ligands (Bcl11 is a target of Notch1 signaling) [132,133]. Further studies in *Vav1-Cre-Rbp-JK^fl/fl^* mice showed that not only is the number of early BM NKPs reduced, but the maturation of splenic NK cells is also blocked [129,130,131]. Rbp-JK is an essential transcriptional effector in the Notch signaling pathway [113]. The perturbed NK cell development may result from the significantly reduced expression of Ets-1 and Ikaros in NKPs of Rbp-JK-deficient mice [129,130,131]. In addition, activated Notch signaling can increase the cytotoxic effect of NK cells by directly enhancing KIR expression [134]. Similarly, the expansion and differentiation of human CD3ε^−^CD56^+^ NK cells from CD34^+^ HSC is accelerated in the presence of Jagged-2 and Delta-1 and -4, suggesting the importance of Notch signaling [135]. These studies demonstrate that Notch boosts NK cell differentiation; however, it is not essential.

### 4.2. Ikaros Regulates the Commitment of HSCs to Early LMPPs

Ikaros is one of five Ikaros TF family members, which includes Aiolos (also known as Ikaros family zinc finger protein 3, IKZF3), Helios, Eos, and Pegasus [136]. The expression of these family members is tissue-specific [137,138]. Ikaros, Aiolos, and Helios are enriched in hematopoietic tissue, including NK cells [139,140,141]; Eos and Pegasus are expressed in multiple tissues [136]. Ikaros was characterized as a master regulator for lymphoid lineage specification [142], which is expressed in HSC, LMPP, CLP, T, B, and NK cells [143]. As a zinc finger protein, Ikaros has four N-terminus DNA binding domains and two C-terminus dimerization domains [144,145]. Alternative splicing of *Ikzf1* generates multiple Ikaros isoforms, which all share a conserved transcriptional activation domain and a dimerization domain [146]. The homo- and hetero-dimers of DNA-binding isoforms inhibit or activate transcription by regulating their DNA binding affinity [146]. The isoforms that lack DNA-binding domains dimerize with other family members, act in a dominant-negative fashion by sequestering them away from DNA, and regulate gene transcription [146]. Ikaros null mutation mice are generated by the deletion of the C-terminal of *Ikzf1,* which encodes the activation, dimerization, and other protein interaction domains, resulting in the lack of HSCs, B, and NK cell lineages, but maintaining part of the T cell differentiation potential [140,147]. Ikaros-deficient mice lack HSCs, which leads to the loss of NK cells either in the spleen [140,148] or an in vitro system where NKPs cannot differentiate into mature NK cells [149]. There is a complete loss of Flt3 expression and reduced level of c-Kit in Ikaros-deficient lymphoid progenitors, which are two critical cytokine receptors to drive HSCs to CLPs [140,148]. ChIP-seq data show that c-Kit is mostly downregulated in progenitor B cells from Ikaros-deficient mice compared to WT mice [150,151]. The regulation of Flt3 and c-Kit expression may partially explain the NK cell defect in these mice, although there is no direct evidence to show that Flt3 is the target gene of Ikaros. Both Aiolos and Helios are expressed in NK cells [152,153]. Aiolos and Helios help in NK cell maturation and function, respectively [139,152].

#### 4.2.1. Aiolos Regulates NK Cell Terminal Maturation

Aiolos has been reported for its notable expression in human NK cells, especially higher in CD56^dim^ compared to CD56^bright^ NK cells [153]. Aiolos-deficient mice have a higher number of CD11b^−^CD27^+^ and CD11b^+^CD27^+^ NK cells; but, less CD11b^+^CD27^−^ population [139]. The high proliferation in response IL-15 in Aiolos-deficient mice indicates it blocks the differentiation before the mature stage, instead of impairment in proliferation [139]. Aiolos is an indispensable TF for NK cells to clear tumor cells and to produce IFN-γ [139]. The deletion of the DNA-binding domain of Ikaros results in an active dominant-negative form of Ikaros. These mutant mice exhibit a more severe phenotype than Ikaros null mice due to the total lack of T, B, NK, and DC lineages, and their earliest defined progenitors, and do not survive more than three weeks after birth [137].

#### 4.2.2. Helios Regulates NK cell Effector Functions

Helios was identified as a T cell activation marker and played an important role in effector T regulatory cells [154,155]. Recently, it was reported that a hyperresponsive phenotype of NK cells caused by N-ethyl-N-nitrosourea (ENU) mutation downregulates Helios in NK1.1^+^CD11b^+^ compared to the NK1.1^+^CD11b^−^ population [152]. The downregulation of Helios in the NK1.1^+^CD11b^+^ population impairs effector function potentials of NK cells such as IFN-γ production and cytotoxicity against YAC-1 via NKp46 [152]. This observation suggests that Helios may be able to modify the threshold of reactivity of NK cells as it was reported in B cells [156]. ChIP assays demonstrated that Helios binds to the FoxP3 promoter region, and knocking-down Helios causes the downregulation of FoxP3 in T cells [157]. Previous studies showed that FoxP3 suppresses immune function not only in T cells, but also in FoxP3-modified human NK cell lines through immunosuppressive cytokine IL-10 [158]. However, additional studies are warranted to define the mechanism by which Helios regulates the effector functions of NK cells.

### 4.3. Ets-Family TFs

The E-twenty-six (Ets) family are winged helix-turn-helix TFs, which share a common Ets domain that binds to purine-rich DNA motifs with a conserved GGAA sequence [159]. These TFs control a range of divergent biological processes, such as cellular activation, differentiation, and oncogenesis [160,161]. Purine-rich box 1 (PU.1), Ets proto-oncogene 1 (Ets-1), and Myeloid Elf-1 like factor (Mef) are three important TFs in this family, which regulate NK cell development [162,163,164].

#### 4.3.1. PU.1 Regulates the Commitment of CLPs to NKPs

PU.1 plays a role in multiple hematopoietic lineage differentiation [165,166,167]. In PU.1-deficient mice, except few T cells, monocytes, granulocytes, myeloid-derived DCs, and B cells are incapable of developing [168,169]. PU.1 expression starts from the CLP stage [170,171] and can be detected in immature and mature NK cells [163]. PU.1-deficient mice die either at late embryonic life or after birth before NK cell transit from the prenatal liver into the periphery [168,169]. To explore the effect of PU.1 on NK cells, hematopoietic chimeras were used [163]. Cells from WT or PU.1-deficient fetal livers were injected into irradiated *Rag2^−/−^*γ_c_*^−/−^* mice. NK cells are present in the BM, spleen, and liver of PU.1-deficient chimeras, albeit with reduced NKP and iNK cell numbers [163], suggesting that PU.1 plays an intrinsic role in NK lineage commitment from CLP to NKP, but is not essential. PU.1-deficient mice have a lower fraction of NK cells in interphase (G_1_, S, and G_2_), suggesting PU.1 positively regulates proliferation [163].

PU.1-deficient mice are unable to drive NK cell expansion and activation in response to the combination of IL-2, SCF, Flt3, and IL-7 [163]. This suggests that PU.1 regulates NK cell development through cytokine signaling. The downregulated expression of c-Kit in PU.1-deficient NK cells may partly explain the developmental defects [163]. ChIP-seq data show that PU.1 represses Ets-1 expression in T cells [172]. Ets-1 is upregulated in *PU.1^−/−^* chimeras of NK cells [163], which suggests that Ets-1 compensates for lack of PU.1. In addition, the interaction with Runx1 is required for PU.1 to bind DNA motif and regulate gene expression [172]. Another downstream target of PU.1 is Kruppel-like factor 4 (Klf4) that is a key TF in the regulation of stem cell pluripotency [173]. Splenic and blood mNK cell numbers are significantly reduced in *Mx1-Cre-Klf4^fl/fl^* mice; however, the number of NK cells in the BM and liver is not altered [174]. The reduced NK cells can be rescued when Klf4-deficient NK cells are transferred to WT mice [174]. These suggest that PU.1 may extrinsically regulate specific tissue environment through Klf4 to regulate NK cell development.

The proliferation of NK cells from PU.1-deficient mice are dampened in response to IL-2 [164]. However, these NK cells are able to lyse the sensitive target cell line, YAC-1, even though the c-Kit expression level is reduced [163]. Splenic NK cells from PU.1-deficient chimeras have comparable expression levels of DX5, Ly49C, Ly49I, and Ly49G2, but selectively reduced Ly49A and Ly49D [163]. Ly49D is a DAP12-associated activating receptor. DAP12 is downregulated in PU.1-deficient myeloid cells [175,176]. PU.1-deficient NK cells produce less IFN-γ due to the downregulated DAP12-dependent signaling pathway. However, the DAP12 signaling pathway has not been examined in PU.1-deficient NK cells. Earlier work also showed that PU.1 plays a critical role in upregulating the IL-12 signaling pathway in a macrophage cell line [177]. We predict a similar role of PU.1 in NK cell-mediated activation via IL-12R.

#### 4.3.2. Ets-1 Regulates the Transition of NKPs to iNKs

Ets-1 plays a central role in the development and survival of NK cells, the survival of T cells, and the terminal differentiation of B cells [162,178,179,180]. NK cells start expressing Ets-1 as early as in the LMPP stage, reach the peak at the late stage of NKP, and maintain the high expression level until the mNK stage [181]. The *Ets-1^−/−^* mice data show that Ets-1 is responsible for the optimal development of 50% of iNK and about 80–90% of mNK cells in the BM and spleen [181]. NKP numbers are not affected in Ets-1-deficient mice [181]. This suggests that the development of NK cells is stopped at the NKP stage in *Ets-1^−/−^* mice. In addition to being identified as a key TF in the early development of NK cells, Ets-1 has also been shown as a critical TF during post-NKP stage development in humans [182]. The *NKp46-Cre-Ets-1^fl/fl^* mouse model was generated by conditionally deleting *Ets-1* exclusively at the beginning of the immature NK stage, utilizing the promoter of the *NKp46* gene, which is expressed during the late immature stage of murine NK cells [183]. Total NK cell numbers in *NKp46-Cre-Ets-1^fl/fl^* mice are significantly reduced compared to *Ets-1^fl/fl^* or *NKp46-Cre* in the BM, spleen, and blood, suggesting that Ets-1 plays a role in the post-iNK stage.

*Ets-1^−/−^* mice display similar phenotypic impairments to that of mice lacking IL-15 or γ_c_ chain. Production of IL-2, IL-15, and the expression of their receptors in mNK cells in *Ets-1^−/−^* mice are comparable to that of WT mice [162]. Notably, stimulation of human NK cells with IL-2 and IL-15 activates the expression of ETS-1 [184]. This activation process depends on the MEK1/ERK1 pathway [184]. Jak3 and Stat5, two critical downstream components of the IL-15 signaling pathway, have been shown to be closely related to Ets-1. Stat5 and Ets-1 proteins form a complex to activate gene transcription in T cells [185]. Ets-1 can directly bind the *Jak3* promoter region to activate transcription [186]. The close relationship to the IL-15 signaling pathway may explain the developmental defects of NK cells in Ets-1-deficient mice. In addition, Ets-1 directly binds to the *Idb2* (Id2 gene) promoter region and activates its transcription in mNK cells [181]. Further evidence shows that Ets-1 also promotes the expression of T-bet [181]. Both Id2 and T-bet are crucial TFs in NK cell development and either decreased Id2 or T-bet results in less mature NK cells [187].

Residual mNK cells from *Ets^−/−^* mice are unable to clear co-cultured target lymphoma cell lines, such as YAC-1 and RMA-S [162]. Degranulation triggered by activating receptors is significantly decreased among residual mNK cells of *Ets^−/−^* mice [181]. However, IFN-γ production is not significantly affected [181]. Loss of activating receptors, including NKp46, Ly49D, and Ly49H in Ets-1-deficient mNK, may explain these impairments [181]. Ets-1 limits cytokine-driven NK cell activation, as shown in *Ets^−/−^* mice, which have the considerable upregulation of *Nfil3*, *Gzmb*, and *Prf1* mRNA level and CD69 expression [181]. IL-2 and IL-15 and their receptors are expressed at normal levels [162]. However, exogenous stimulation of IL-2 or IL-15 failed to rescue the defects in cytotoxicity [162]. A further detailed study is required to gain better insights into the role of Ets-1 in NK cell development.

#### 4.3.3. Mef Regulates the Maturation of NK Cells

Mef, encoded by *Ekf4*, is detected in both myeloid and lymphoid lineages [188,189,190]. Mef-deficient mice have normal T and B cell development; however, they have only 40% of NK and 30% of NKT cells in their spleen [164]. These NK cells are defective in mediating effector functions. We predict Mef may plays a role in the maturation of NK cells due to the reduced number of the NK1.1^+^ population and the dampened expression of DX5 in *Mef*-deficient mice [164]. However, direct evidence is required to precisely define the stage where NK cell development is blocked in these mice. Expression levels of IL-15R complex are unaltered in the splenic NK cells of *Mef*-deficient mice [164].

Like PU.1, Mef is also not obligatory for NK cell development because of the detectable number of peripheral NK cells. However, the splenic NK cells from *Mef*-deficient mice produce a lower level of perforin, are unable to lyse the target tumor cells, and secrete less IFN-γ [164]. Components of the IL-2R complex are expressed at normal levels in *Mef*-deficient NK cells; however, the addition of exogenous IL-2 failed to rescue their effector functions [164]. Thus, the IL-2 signaling pathway is not able to explain the defective NK cell effector functions. Further, Mef binds to the Ets-1-binding region and proximal *Perf1* promoter region, which positively regulates perforin expression [164]. The loss of this positive regulation partly explains the defective effector functions in *Mef*-deficient NK cells.

### 4.4. Runx3 Regulates the Commitment to iNK Stage

Runx3 is one of Runx family TFs [191]. They contain a conserved Runt domain that binds to the DNA sequence and heterodimerizes with the binding partner CBFβ (core binding factor) [191]. Binding with CBFβ can both increase the affinity for shared binding sequences and activate or repress target gene transcription [192,193,194]. In mice, Runx1 is primarily expressed in CD4^+^ T cells [195,196]; Runx1 and Runx3 have similar expression levels in NKT cells [195] and Runx3 is predominantly expressed in CD8^+^ T and NK cells [195]. Runx3 expression starts from NKPs and is particularly enriched in iNK and mNK cells [195]. Runx3 is expressed as early as the NKP stage; however, its role in the early developmental stages is yet to be determined. Transplanting Runx3-defective HSCs from transgenic mice expressing the dominant-negative form of Runx3 into *Rag2^−/−^* mice resulted in significantly less CD122^+^ iNK and mNK cells [195]. Runx3 controls the phenotype of NKPs by binding to the CD122-encoding *Il2rb* gene promoter region and initiating transcription in both human and mice [195]. Lack of Runt domain-binding partner, CBFβ, results in the blockade of NK cell development at the pre-NKP stage [197]. Furthermore, the mature CD27^−^CD11b^+^ NK cell percentage and absolute cell numbers are reduced in Runx3-defective mice [195,198]. The expression of the mature NK cell markers, such as CD11b, Ly49C, Ly49I, Ly49F, Ly49G2, and Ly49D, decreases in the absence of Runx3 [195]. Similarly, Runx3 promotes the expression of NK cell-specific receptors in human NK cells [195]. Runx3 binds to the promoter regions of *KIR* and *NKp46* genes and activates their expression in human NK cells [199,200]. This suggests that Runx3 plays a role in NK cell maturation. Future experiments such as BM chimeras using conditionally-deleted CBFβ or individual Runx proteins in NK cells are required to determine their role in each stage of NK cell development.

Since *Il2rb* is one of the Runx targets in NK cells [195], in its absence IL-15-mediated NK cell proliferation is impaired [198]. ChIP-seq results revealed that Runx3, cooperating with Ets-1 and T-box TFs, promotes IL-2/IL-15-mediated NK cell proliferation by activating transcription of *Tnfrsf9* and *Styk1* [198,201,202,203]. In both human and murine NK cells, Runx3 occupancy regions are significantly associated with proliferation, maturation, and migration [198,204]. *Runx3*-deficient NK cells produce comparable amounts of perforin and granzymes and have a comparable ability to lyse YAC-1 cells to that of WT [195,198]. The expression of NKG2D, a major receptor for YAC-1 recognition, is also comparable [195]. With the IL-2 and IL-12 stimulation, these NK cells enhanced the production of IFN-γ, although they were of a relatively immature phenotype [195]. These results indicate that although it regulates the expression of multiple mature markers, Runx3 plays a moderate role in regulating effector functions of NK cells.

### 4.5. E4BP4 Regulates the Transition from NKP to iNK

E4BP4 (E4-binding protein), encoded by the *Nfil3* (nuclear factor interleukin-3) gene, belongs to the basic leucine zipper TF family. It is highly expressed in NK and NKT cells, but not in T or B cells [205]. The expression of E4bp4 starts as early as the CLP stage [206]. However, it is highly expressed in the iNK and mNK stages [205]. Using the *Nfil3^−/−^* mice model, E4bp4 has been shown as a crucial TF during the transition of NKP to the iNK stage [205]. These mice have lower NK cell numbers in the BM and almost undetectable NK cells in the blood or spleen. The CD122^+^NK1.1^−^CD3ε^−^ NKP populations from *Nfil3^−/−^* and WT BM have no substantial differences [205]. However, the CD122^+^NK1.1^+^CD11b^−^ iNK cells are considerably less in the *Nfil3^−/−^* BM, and near-complete loss of CD122^+^NK1.1^+^CD11b^+^ mNK cells [205]. Thus, the high expression of E4bp4 coincides with the transition into iNK and mNK cells. Contrary to these findings, Brady et al. reported that both the percentage of NKPs and their absolute numbers are also significantly decreased in *Nfil3^−/−^* BM cells [206]. Few residual NK cells in the *Nfil3^−/−^* mice produce lower levels of IFN-γ and mediate defective cytolytic functions [205,207]. *Nfil3^−/−^* BM transplant chimeras could not rescue this defect, indicating that these developmental defects are NK-cell-intrinsic [207].

Do the developmental defects in *Nfil3^−/−^* mice involve cytokine signaling pathways? Expression of E4bp4 is not detectable in cultured murine NK cells without IL-15 [54]. Upon the exogenous IL-15 stimulation, E4bp4 expression is upregulated [54]. Furthermore, earlier studies show that both *Pdk1^−/−^* and *Il15rα^−/−^* mice display a significantly lower level of CD122^+^NK1.1^+^CD11b^−^ iNK population [54,74,76]. PDK1 is an essential metabolic regulator connecting PI(3)K to downstream mTOR complexes. Overexpressing E4bp4 rescued NK cell developmental defects in both *Pdk1^−/−^* and *Il15rα^−/−^* mice [54,205]. This suggests E4bp4 is a downstream target of IL-15, and its expression depends on the IL-15-PI(3)K-PDK1-mTORC1-E4bp4 signaling pathway [54]. Except for a few, most of the targets of E4bp4 in this signaling pathway are yet to be identified. E4bp4 can directly bind to the *Eomes* and *Id2* promoter region and promote their expression [206]. Eomes and Id2 are critical TFs for the late-stage development of NK cells. Overexpression of Id2 in *Nfil3^−/−^* NK cells rescued its few numbers of NK1.1^+^CD122^+^ NK cells [205]. These suggest that Eomes and Id2 are some of the essential transcription factors downstream of E4bp4, directing NK cell development [206]. E4BP4-mediated NK cell differentiation and anti-tumor effector function are enhanced in *Smad3^−/−^* mice [208]. The potential mechanism is the regulation of Smad3 in Nfil3, Id2, and IRF2 [208,209]. Comprehensive analyses of *Nfil3^−/−^* BM NK cell developmental data show that they have lower expression of NKp46, NKG2D, DX5, and CD43, which are the critical activating NK cell receptors. In contrast, the expression of inhibitory receptors such as Ly49C, Ly49I, Ly49F, and KLRG1 is higher [207]. Changes in these receptors lead to refractory NK cells, which may explain their defective effector functions.

### 4.6. ID2 and E Proteins Regulate iNK to mNK Transition

Inhibitors of DNA-binding (Id) proteins, a family of helix-loop-helix (HLH) TFs including Id1 through Id4, are transcriptional repressors. They operate by heterodimerizing with E proteins and preventing them from binding to DNA [210]. E proteins, which include E12, E47, HEB, and E2-2, either form homodimers as transcriptional activators or heterodimers with class II basic HLH TFs as both transcriptional activators or repressors and regulate T or B cell development [211,212]. A functional balance between E and Id proteins controls the lymphocyte differentiation programs [212,213,214]. Id3 and Id2 were identified as molecular switches in lymphocytes lineage specification. Id2 is the predominant Id protein expressed in NK cells [187,215,216]. The constitutive expression of ID2 or ID3 enhances the differentiation of CD34^+^ cells into NK cells; however, it blocks the commitment of CD34^+^ cells into T lineage in fetal thymic organ cultures (FTOCs) [215,216]. T cell developmental defects result from the arrested HEB DNA-binding by Id2. HEB is the only E protein favoring T lineage commitment and inhibiting NK cell development [216]. While the high level of Id2 synergizes with IL-15 to promote human NK cell development by increasing the numbers of NKPs [216]. Further studies show that the number of CD122^+^NK1.1^+^CD49b^+^ mNK cells in the BM of *Id2^−/−^* mice significantly decreases, but the numbers of NKPs and iNK cells are normal [187]. This suggests Id2 is not an essential TF for early NK cell development in the BM, and lack of Id2 causes a blockade at the post iNK and pre-mNK cell stages in BM. Transplantation of BM from *Id2^−/−^* mice to WT recipients resulted in sub-optimal NK cell differentiation, indicating that this defect is NK-cell-intrinsic [217]. Similarly, among *Id2^−/−^* splenocytes, both total NK and mNK cell numbers are reduced, although the development and functions of T and B cells are normal [217]. Interestingly, the majority of *Id2^−/−^* mNKs in the spleen express high IL-7Rα^+^, which is a marker for thymus-derived NK cells [217]. This suggests that the splenic NK cells in *Id2^−/−^* mice are predominantly thymus-derived. In vitro culture experiments showed that NK cells from the *Id2^−/−^* BM could not differentiate further in response to IL-15 stimulation [217]. This demonstrates that Id2 is downstream of IL-15. Id2 is required for the thymic NK cells to mature and obtain their distinct phenotype [218].

As is characteristic of the Id protein family, Id2 acts as a repressor by binding and sequestering E proteins [210]. Loss of *E2A*, which encodes E12 and E47, rescues NK cell differentiation in the BM of *Id2^−/−^* mice [187,216], but failed to rescue the defective mNK transition in the *Id2^−/−^* spleen [187]. These different phenomena between the spleen and BM of *Id2^−/−^* mice strongly indicate distinct requirements. Id2 also binds to other TFs, such as Ets-1, E4bp4, and retinoblastoma (Rb). Interaction of Id2 and Rb drives cell proliferation [219], which may explain the reduced NK cell numbers in the BM and spleen of *Id2^−/−^* mice. Id2 is the direct downstream target of Ets-1 and E4bp4. Few remaining *Id2^−/−^* NK cells are capable of killing YAC-1 cells, but with a reduced efficiency [187,217]. They also produce less IFN-γ in response to IL-2 and IL-12-mediated stimulation [187,217].

### 4.7. TOX is Essential for the Transition of mNK to Terminally Mature NK Cells

The TOX (thymocyte-selection associated high mobility group box) family contains a conserved DNA binding HMG-box motif and is initially identified as an essential TF in T cell selection and CD4^+^ T cell development [220,221]. There are four members in the Tox TF family. TOX1 and TOX2 are expressed in hematopoietic cells [222,223]. However, TOX1 and TOX2 have distinct expression patterns during human umbilical cord blood (UCB) NK cell differentiation in vitro [224]. TOX1 is expressed as early as HSC and maintains high expression at the iNK stage. TOX2 starts low expression from the HSC stage and a continuous high expression level until the transitional NK stage [224].

NK cells from both spleen and BM of *Tox1^−/−^* mice have a comparable number of NK1.1^−^DX5^−^ NKPs, and NK1.1^+^DX5^−^ iNKs, but do not possess any DX5^+^ mNK cells [225], suggesting that NK cell development in *Tox1^−/−^* mice are blocked at the post-iNK stage [225]. This developmental defect is NK-cell-intrinsic since the transplant of *Tox1*-deficient NK cells into WT mice could not rescue the defects [225]. In line with the finding in *Tox1^−/−^* mice, the Lentivirus-mediated *TOX*-knockdown of human UCB NK cells shows a blockade at the CD117^+^CD94^−^CD56^−^ transitional NK cell stage [224]. Overexpression of TOX2 can enhance NK cell development and increase the number of mature NK cells [224]. Further, ChIP results show that TOX2 directly binds the *Tbx-21* (T-bet) promoter region [224]. T-bet expression in *TOX2* knockdown NK cells is substantially decreased in transitional and mature NK cell stages, but the expressions of TOX1, ID2, or NFIL3 are unaltered [224]. Overexpression of *Tbx21* can rescue the in vitro NK cell developmental defects in *TOX2* knockdown cells [224], suggesting that TOX2 regulates human NK cell development through T-bet. The knockdown of *TOX2* reduced the degranulation and perforin expression when NK cells were co-cultured with K562 cells [224]. The reduced expression of activating receptors, such as NKp46 (NCR1) and NKp30 (NCR3), in these knockdown NK cells may partially explain the defective effector functions [224]. The role of Tox proteins in cytokines-mediated activation pathways are yet to be determined.

### 4.8. IRF1 and IRF2 Regulate the Transition of iNK to mNK Stage

The interferon regulatory factor (IRF) family members are identified for their binding ability to promoters of IFN-α, -β genes, and IFN-stimulated regulatory elements [226,227]. IRF-1- or IRF-2-deficient mice exhibit severe NK cell defects [228,229,230,231]. A set of BM chimera experiments showed that IRF-1 is required for IL-15 production by stromal cells within the BM microenvironment. However, the defects in IRF-2-deficient NK cells are cell-intrinsic. Irrespective of their normal proliferation, the total numbers of NK cells in both BM and the periphery of *Irf2^−/−^* mice are significantly reduced due to accelerated apoptosis [230,231]. Among the remaining NK cells in the spleen and liver, the percentage of CD11b^+^DX5^+^ or CD11b^+^CD43^+^ mNK cells is significantly lower in *Irf2^−/−^* compared to WT mice, but the percentage of iNK cells is similar to that of WT mice [230,231]. The few remaining NK cells are capable of lysing YAC-1 cells and produce moderately less IFN-γ in response to IL-12. Upstream signaling molecules and targets of IRF-2 that are involved in NK cell development have not been defined.

### 4.9. FoxO1 Suppresses NK Cell Lineage Commitment and NK Cell Maturation

Forkhead transcription factors of the O class (FoxO) family contains a winged-helix DNA-binding domain and the forkhead domain [232]. There are four members in this family: FoxO1, FoxO3, FoxO4, and FoxO6. FoxO1 is highly expressed in NKPs and iNKs compared to mNKs. FoxO3 keeps a relatively low expression profile throughout the NK cell development [233]. FoxO1 and FoxO3 redundantly suppress NK cell maturation. Both *Ncr1-Cre -FoxO1^fl/fl^* and -*FoxO3^fl/fl^* mice possess larger CD27^−^CD11b^+^ mNK, a moderately increased CD27^+^CD11b^+^, and comparable CD27^+^CD11b^−^ iNK cell populations [234]. However, double knockout of *FoxO1* and *FoxO3* did not increase mature NK cell populations compared to single knockouts [234]. Thus, FoxO1 plays a dominant role in the early stage of NK cell development. FoxO3, with considerably less expression, has relatively less influence on the mature NK cell population [234].

However, Ncr1-driven *FoxO1* deletion failed to define the role of FoxO1 in the earlier developmental stages. A hematopoietic specific deletion in *Vav1-Cre-FoxO1^fl/fl^* mice that covers the early developmental stages of NK cells shows an increased ratio of CLPs (Lin^−^c-Kit^+^Sca-1^+^CD127^+^), NKPs (Lin^−^CD127^+^CD122^+^), and mNKs (CD27^−^CD11b^+^), and a decreased percentage of iNKs (CD27^+^CD11b^−^) [234]. FoxO1 suppresses NK cell maturation by directly inhibiting the transcription of T-bet [234]. FoxO1 is downstream of mTORC2. Work from our laboratory reported that FoxO1 suppresses the transition of iNK to mNK cells through the axis of mTOR2-Akt^S473^-FoxO1-T-bet [235]. FoxO1 can suppress the proliferation of NK cells by directly promoting transcription of genes that encode cell-cycle inhibitors [234]. So far, it is not clear how FoxO1 plays a role in mNKs. When NK cells are challenged with tumor or cytokines, FoxO1 is phosphorylated into an inactivated form and translocated into the cytoplasm for degradation, which leads to the activation of NK cells [234]. In contrast, loss of FoxO1 (*Ncr1-Cre-FoxO1^fl/fl^*) enhances NK cell cytotoxicity and IFN-γ production in response to YAC-1 or cytokine-mediated stimulation [234]. Recent work shows that FoxO1 severely impairs the late stage of NKT cell development [236].

### 4.10. Eomes and T-Bet Antagonize Each Other to Regulate NK Cell Maturation and Terminal Maturation

Eomesodermin (Eomes) and T-bet are two members of the T-box family of TFs, which control multiple aspects of NK cell development and maturation [237,238,239]. Both Eomes and T-bet contain a highly conserved DNA-binding domain that indicates that they bind to the same DNA motif. However, a variable C-terminal domain between Eomes and T-bet suggests their interacting partner and biological functions are different [240,241]. NK cell development in *Tbx21^−^^/^^−^* (T-bet knockout) mice is blocked before the terminal NK cell maturation stage. Lack of T-bet decreased the expression of CD11b, DX5, KLRG1, and CD43, while increasing the immature marker c-Kit [237,238]. However, the level of Ly49 receptors is not significantly affected [242]. The accumulation of NK cells in the BM and lymphoid tissue of *Tbx21^−^^/^^−^* mice may result from the decreased expression of sphingosine-1-phosphate receptor 5 (S1P5) [243], which is an indispensable factor for NK cell egress from lymphoid and BM niches in both mouse and human models [244,245]. The reduced NK cell numbers are unable to be rescued when *Tbx21^−^^/^^−^* BM is transplanted into a WT recipient [237,238]. Homozygous deletion of Eomes is lethal as mice die during early embryogenesis [246]. Therefore, the role of Eomes has been studied on *Vav1-cre-Eomes^fl/fl^* mice [237]. In the absence of Eomes, the total NK1.1^+^ or NKp46^+^ NK cell population is decreased [237]. The development is blocked in the immature stage as they are unable to transit into the CD27^+^CD11b^+^ stage. NK cells from these mice have significantly decreased expression of DX5 and Ly49 receptors and increased expression of TRAIL (an immature NK cell surface marker) [237].

Eomes and T-bet expression largely overlap in CD8^+^ T cells and NK cells [237,238,239]. Eomes and T-bet have been reported to play a redundant role in the differentiation of CD8^+^ effector T cells and supporting the role in specifying the fate of CD8^+^ T cells [239,247,248,249]. T-bet and Eomes antagonize the expression of each other, because T-bet-deficient mice express a higher level of Eomes in NK cells and vice versa [237,238]. Interestingly, the absence of one Eomes allele in *Tbx21^−^^/^^−^* mice more severely decreased NK cells in the periphery than in *Tbx21^−^^/^^−^* mice [239]. Total loss of Eomes and T-bet (*Vav1-cre-Eomes^fl/fl^ Tbx21^−^^/^^−^*) results in the absence of NK1.1^+^ or NKp46^+^ cell population in both the BM and periphery [237]. Results from a single or double loss of Eomes and T-bet suggest that the cooperation of T-bet and Eomes is essential for NK cell development, and they share several functions [237]. Both T-bet and Eomes are regulated by IL-15 and mTORC signaling [250]. mTOR complexes can negatively regulate the expression of T-bet and suppress NK cell maturation [250]. The T-bet expression can be upregulated by IL-12 and IL-15 stimulation [251,252]. ChIP-seq results show that T-bet and Eomes are direct targets of Stat5, which is a critical downstream component of the IL-15R signaling pathway [253]. In turn, T-bet and Eomes can cooperatively bind to the *Il2rb* (IL-2Rβ) promoter region to activate transcription IL-2Rβ. Because IL-2Rβ is an essential subunit for IL-2R and IL-15R signaling pathway, Eomes and T-bet play an essential role in NK cell development. Irrespective of these understandings, it is not clear how T-bet and Eomes redundantly or cooperatively regulate NK cell development and functions.

The role of Eomes and T-bet in NK cell effector functions is emerging. Runx3 [198], E4bp4 [206,254], Ets-1 [181], and Tox2 [224] support the expression of T-bet. However, FoxO1 directly binds to the *T-bet* promoter region and inhibits its transcription [234]. T-bet directly binds to the *ifng* promoter region and helps in its transcription [255,256]. The production of IFN-γ in T-bet-deficient NK cells is significantly decreased, but not completely lost [237,238]. T-bet-deficient NK cells can initiate the production of IFN-γ at the early stage of stimulation, but they cannot maintain the productivity and keep the high level of IFN-γ after 24 h stimulation by IL-12 and IL-18 [237,238,257]. This suggests that the early phase of IFN-γ production is T-bet independent, but the maintenance of production depends on normal T-bet function [237,238]. In addition, T-bet is able to bind to *Gzmb*, *prf1*, and *Runx1* promoter regions in NK cells [237,238]. In line with this, T-bet-deficient NK cells display a modest decrease in the ability to lyse YAC-1 target cells and decreased expression of perforin [237,238,239]. Eomes can cooperate with T-bet to augment cytolytic effects because the expression of perforin shows a severe downregulation in *Tbx21^−^^/^^−^Eomes^+/^^−^* NK cells compared to *Tbx21^−^^/^^−^Eomes^+/+^* NK cells [239]. Similarly, IFN-γ production is also significantly decreased in *Tbx21^−^^/^^−^Eomes^+/^^−^* CD8^+^ T cells compared to *Tbx21^−^^/^^−^Eomes^+/+^* cells [239]. Indeed, Eomes can activate the transcription of *Gzmb* and *Prf1* and control the expression of Ly49 receptors [237,239,258]. However, Gordon at al. reported that NK cells from *Vav1-cre-Eomes^fl/fl^ T-bet^−^^/^^−^* are able to clear YAC-1 cells [237].

### 4.11. GATAs

GATA family proteins contain two essential Cys4-type zinc fingers. The C-terminal zinc finger domain is involved in most of the DNA-binding activity. In contrast, the N-terminal zinc finger domains bind unique DNA motifs and interact with important transcriptional cofactors. GATA1, GATA2, and GATA3 play essential roles in hematopoietic cells. GATA2 and GATA3 play important roles in NK cell development and functions.

#### 4.11.1. GATA2 Regulates Human iNK Transition to CD56^bright^ Cell

GATA2 deficiency syndrome displays variable clinical manifestations [259,260], but the most reliable of them are B and NK cell lymphopenia and monocytopenia [261]. A recent study, including 57 patients with GATA2 deficiency, showed that 70% of them experienced severe viral infections, 35% of them diagnosed with HPV- or EBV-associated tumors [260]. These indicate the defective NK cell-mediated anti-viral and anti-tumor functions. NK cell deficiency among these patients displays severely reduced numbers of peripheral NK cells, a specific loss of the CD56^bright^ subset, and defective NK cell cytotoxicity in remaining CD56^dim^ NK cells [262]. The fact the exclusive expression of GATA2 in CD56^bright^ cells (15-fold increase compared to CD56^dim^) [262] may explain the importance of GATA2 in maintaining the CD56^bright^ subset. *Gata2* knockout mice died at the embryonic stage due to severe hematopoietic defects. The NK cell lineage-specific *Gata2* knockout model is important in order to understand its role in NK cell biology.

#### 4.11.2. GATA3 Regulates the Maturation of iNK and thymic NK Cells

Gata3 is dispensable for conventional NK cell development. Splenic NK cells from *Gata3*-deficient chimeras (Gata3 deficiency from HSC stage) have normal cell numbers and expression levels of cell surface markers, whereas they have fewer NK cells in the liver and impaired IFN-γ production [263]. Gata3 is required for thymic NK cell development [64]. Thymic NK cells are distinct from conventional NK cells with a reliance on IL-7R signaling. Recently, a study based on the NK cell-specific *Gata3*-deficient model (*NKp46-Cre-Gata3^fl/fl^*) showed that NK cell development in both the spleen and BM was blocked from the CD27^+^CD11b^−^ the CD27^+^CD11b^+^ stage, indicating that Gata3 promotes NK cell maturation [264]. These *Gata3*-deficient NK cells are unable to exit the BM and to produce IFN-γ, but their ability to clear YAC-1 cells is maintained [264]. The NK cell egress can be rescued following the treatment of CXCR4 antagonists, which recruit iNK from BM cells to blood [264]. Under this condition, CXCR4 antagonist AMD3100 can rescue the defect of *Gata3*-deficient NK cell egression. Previous work showed that the administration of AMD3100 in mice resulted in reduced NK cell numbers in BM and augmented numbers in spleen and peripheral blood [265]. This suggests that the defect of *Gata3*-deficient NK cell egression is associated with the CXCR4 signaling pathway. The impairment of IFN-γ production is partially rescued because of hyporesponsive to cytokine stimulation [264]. Thus, Gata3 is required for NK cell egress and cytokine-mediated IFN-γ production.

## 5. PLZF and Zbtb32 in Adaptive NK Cells

Promyelocytic leukemia zinc finger (PLZF, encoded by *Zbtb16* gene) is a member of the BTB-zinc finger TF family, and is present in both human and mouse NK cells [266,267]. Human adaptive NK cells display significantly downregulated expression of PLZF, which suggests that lack of PLZF marks a gain of human adaptive NK cells [268]. In addition, PLZF is highly associated with markers of liver-resident NK cells in humans [266]. Zbtb32 is another TF from the BTB-zinc finger TF family. The expression of Zbtb32 is induced by murine cytomegalovirus (MCMV) infection, which is required for NK cell expansion following infection [267]. In Zbtb32-deficient mice, the percentage and number of mature NK cells is comparable with WT mice [267]. *Zbtb32^−^^/^^−^* mice display normal phenotypic profiles of NK cells and functional markers, including Ly49H, Ly49D, NK1.1, DX5, Ly49C, Ly49I, and Ly49A [267]. Both WT and *Zbtb32^−^^/^^−^* mice show similar upregulation of granzyme B following murine cytomegalovirus (MCMV) infection [267]. Zbtb32 transcriptionally upregulates IRF8, which is required for the antiviral functions and the virus-driven expansion of NK cells [269].

Group 1 innate lymphoid cells (ILCs) are comprised of NK cells and ILC1s, which share several common features: producing IFN-γ and requiring T-bet for their functions. However, ILC1s are non-cytotoxic, tissue-resident, and non-Eomes dependent ILCs [270,271,272]. common innate lymphoid progenitors (CILPs), differentiated from CLPs, develop into either NKPs or innate lymphoid cell precursors (ILCP) [272,273]. The transcription factors of Gata3 and PLZF play critical roles in the differentiation of ILC1s from conventional NK cells [274,275]. Deletion of PLZF significantly altered the development of ILCs subset, but not NK cell development [273]. Conditional Gata3 knockout in hematopoietic cells (Gata3^fl/fl^-Vav^Cre/+^) showed significantly reduced PLZF^+^ ILCPs [276].

## 6. Summary and Future Outlook

Recent studies have led to significant advancements in the molecular aspects of NK cell biology, including their development and effector functions. With their ability to kill malignant cells without prior sensitization, NK cells possess clinical promise for the formulation of successful immunotherapeutic approaches. However, the inability of autologous NK cells to clear tumor cells in a cancer patients requires an in depth understanding of their transcriptional networks. In addition, a better thorough understanding of the signaling cascades and transcriptional networks operating in human NK cells is obligatory to overcome the immunosuppression that exists in cancer patients. The identity of divergent signaling cascades that exclusively regulate the cytotoxicity and the production of inflammatory cytokines is needed to uniquely target tumor cells with minimal side-effects. Current studies on TFs are performed using murine models and their role in human NK cells are yet to be fully defined. Future work related to the mechanism of the functions of TFs, the co-operation between various TFs, and their regulatory mechanisms should be done to improve NK cell-based immunotherapy.

## Figures and Tables

**Figure 1 cancers-12-01591-f001:**
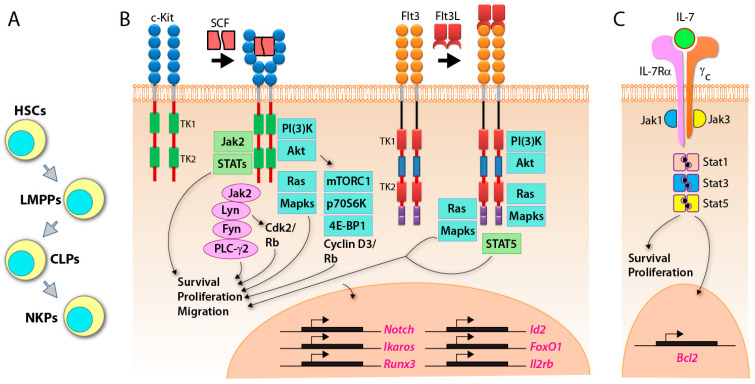
Early programs that commit progenitors to natural killer (NK) cell lineage. (**A**) General schema of hematopoietic stem cells (HSCs)-lymphoid-primed multipotential progenitors (LMPPs)-common lymphoid progenitors (CLPs) to NK cell progenitors (NKPs). (**B**) Stem cell factor (SCF) and FMS-like tyrosine kinase 3 ligand (Flt3L) interact with their tyrosine kinase receptors c-Kit and Flt3L, respectively. The signaling pathways initiated through these receptors, primarily mediated by Jak2-Stats, PI(3)K-mTORC1-S6K1/4EBP1, or Jak2-PLC-γ2-NF-κB, promote cell survival, proliferation, and migration between niches. This activation also leads to essential gene transcriptions including IL-2Rβ. (**C**) IL-7-mediated activation via CD127 results in the maintenance of cell-fate integrity, survival, and proliferation.

**Figure 2 cancers-12-01591-f002:**
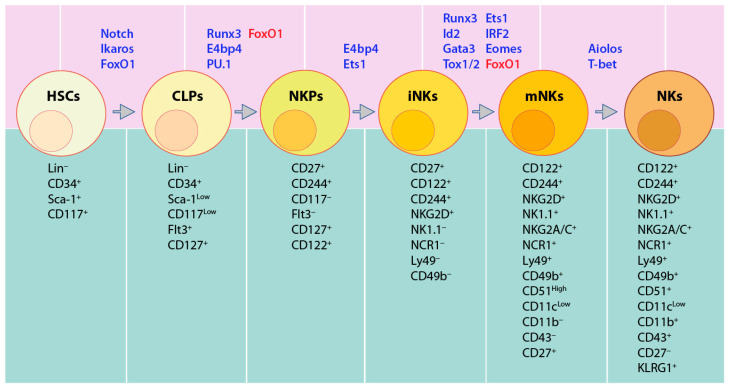
Transcription Factors (TFs) involved in distinct murine NK cell developmental stages. Notch and Ikaros promote Lineage negative (Lin^−^) CD34^+^Sca-1^+^CD117^+^ hematopoietic stem cells (HSCs) committing to common lymphoid progenitors (CLPs, Lin^−^CD34^+^Sca-1^Low^CD117^Low^Flt3^+^), while FoxO1 suppresses this commitment to keep the HSCs in a quiescent state. Runx3, E4bp4, PU.1, and Id2 enhance NK cell lineage transition to CD27^+^CD244^+^CD127^+^CD122^+^ NK cell precursors (NKPs). FoxO1 suppresses NK lineage transition. Committed immature NK cells (iNKs) that are marked by the expression of NKG2D are regulated by E4bp4 and Ets1. Runx3, Ets1, Id2, IRF2, Gata3, Eomes, and Tox1/2 promote the iNK into mature NK cells (mNKs), which are defined by the expression of NCR1, NK1.1, Ly49, CD49b, and CD51. Aiolos and T-bet further enhance mNK to terminal NK cells (tNK) with the expression of CD11b, Ly49s, and KLRG1 and loss of CD27.

**Figure 3 cancers-12-01591-f003:**
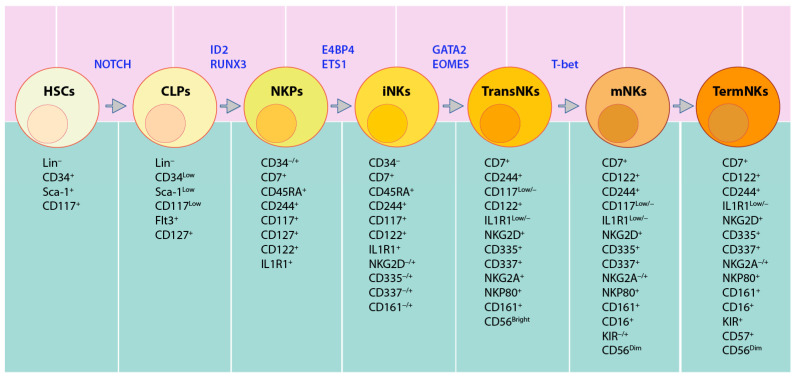
TFs involved in distinct human NK cell developmental stages. Notch proteins facilitate transition from Lin^−^CD34^+^ HSCs to CLPs (defined by Lin^−^CD34^+^CD244^+^). Commitment to NK cell lineage (named NKPs), marked by CD117^+^CD127^+^CD122^+^IL1R1^+^, is activated by Id2 and RUNX3. The transition to iNK is defined by higher expression of IL1R1 and the expression of NKG2D, CD335, CD337, and CD161, which is enhanced by NFIL3 and ETS1. GATA2 and EOMES promote commitment to transitional NK cells (TransNKs) between the immature and the mature stage, which are defined by NKG2D^+^CD335^+^CD337^+^CD161^+^CD56^bright^ population. T-bet promotes them to NKp80^+^CD56^dim^CD16^+^KIR^−/+^ mNKs. The commitment of terminally mature NK cells (TermNKs) is marked by CD56^dim^CD16^+^CD57^+^ KIR^+^.

**Figure 4 cancers-12-01591-f004:**
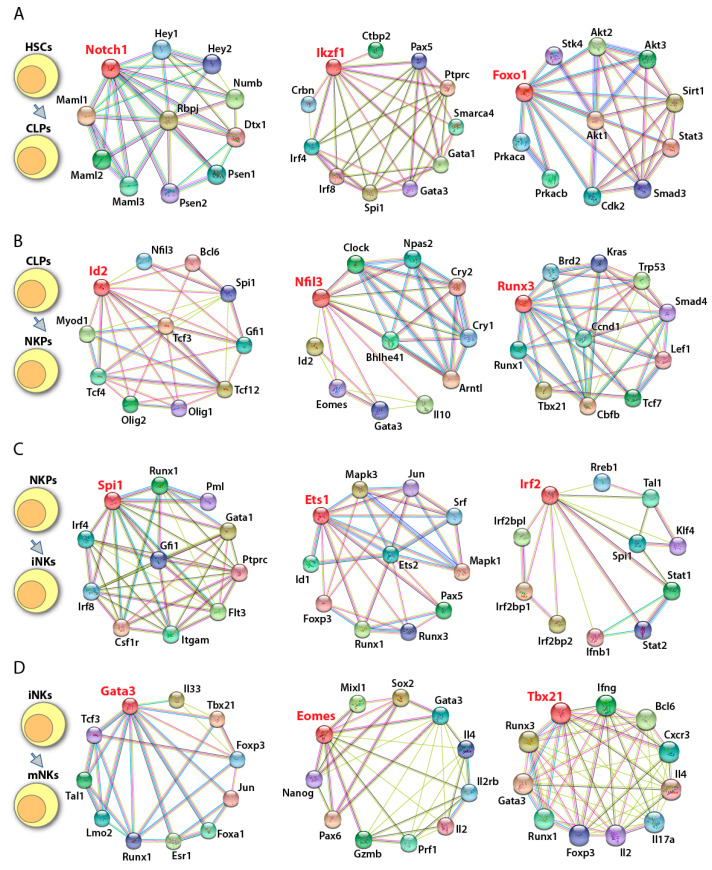
Potential transcriptional networks in murine NK cell development. Potential downstream and upstream regulators of individual transcription factors involved in murine NK cell development were identified using String database (https://string-db.org/). Gene of interest is shown in red font. ‘—’ represent data from curated databases; ‘—’ experimentally determined; ‘—’ gene neighborhood; ‘—’ gene fusions; ‘—’ gene cooccurrence; ‘—’ text mining; ‘—’ co-expression; and ‘—’ protein homology. (**A**) The transition from HSCs to CLPs is regulated by Notch1, Ikaros (encoded by *Ikzf1*), and Foxo1. (**B**) Committed to NK lineage is mainly governed by Id2, Nfil3, and Runx3. (**C**) NKPs transit to immature NK cells under the control of PU.1 (encoded by Spi1), Ets-1, and Irf2. (**D**) The maturation of NK cells is activated by Gata3, Eomes, and T-bet (encoded by *Tbx21*).

**Figure 5 cancers-12-01591-f005:**
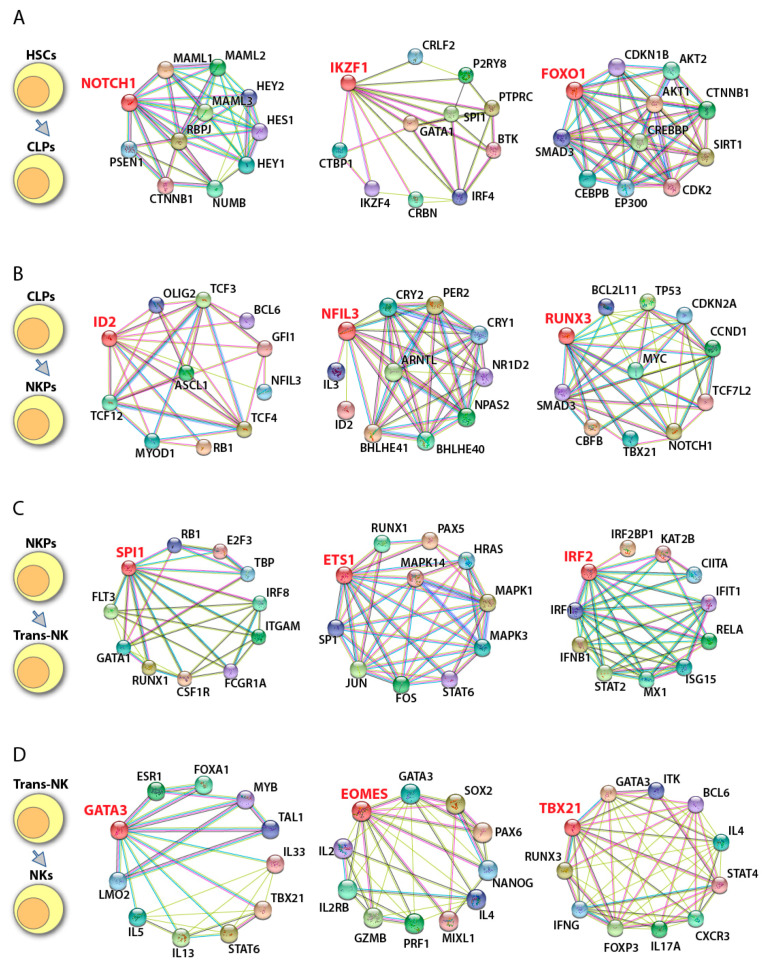
Potential transcriptional networks in human NK cell development. Potential downstream and upstream regulators of individual transcription factors involved in human NK cell development were identified using String database String database (https://string-db.org/). Gene of interest is shown in red font. ‘**—**’ represent data from curated databases; ‘**—**’ experimentally determined; ‘**—**’ gene neighborhood; ‘**—**’ gene fusions; ‘**—**’ gene cooccurrence; ‘**—**’ text mining; ‘**—**’ co-expression; and ‘**—**’ protein homology. (**A**) Human HSCs transit to CLPs under the help of NOTCH1, IKAROS, and FOXO1. (**B**) The commitment to NK lineage is mainly regulated by ID2, NFIL3, and RUNX3. (**C**) Transition from NKPs to iNK is potentially governed by PU.1, ETS-1, and IRF2. (**D**) The maturation of human NK cells is regulated by GATA3, EOMES, and T-bet.

**Figure 6 cancers-12-01591-f006:**
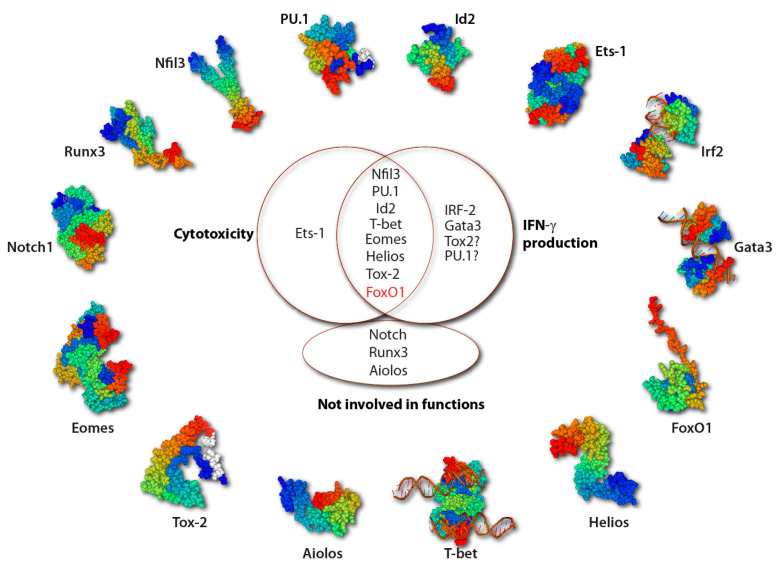
TFs involved in the effector functions of NK cells. The 3-D protein structural motifs of TFs involved in the development and functions of NK cells are shown. Most TFs enhance both cytotoxicity and IFN-γ production of NK cells (black font). However, FoxO1 suppresses both effector functions as a balance (red). Current evidence shows that Ets-1 is involved only in cytotoxicity and IRF-2 and Gata3 are only involved in IFN-γ production. Tox2 and PU.1 may contribute to the IFN-γ production, though no direct evidence exists. Notch, Runx3, and Aiolos are not involved in NK cell effector functions. The ‘?’ in Figure 6 represents no direct evidence showing the effect on IFN-γ production.

**Figure 7 cancers-12-01591-f007:**
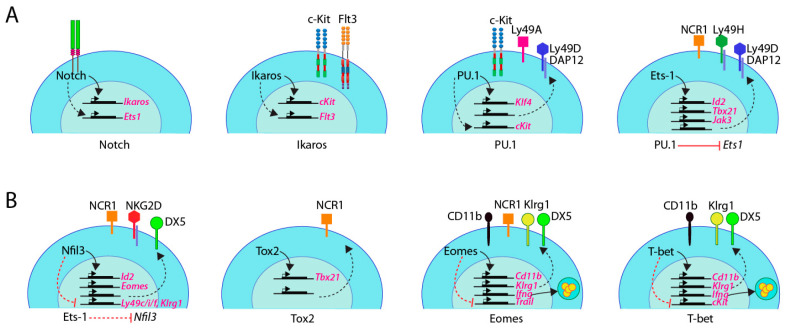
Targets of TFs in murine NK cells. (**A**) Notch proteins regulate NK cell development partially through upregulated expression of Ets1 and Ikaros. Ikaros is indispensable due to its role in the expression of Flt3 and activating the transcription of *cKit* by directly binding to its promoter region. Interaction with Runx1 is required for PU.1 to bind the DNA motif. PU.1 promotes the expression of cKit and Klf4 to contribute to NK cell development. PU.1 upregulates the expression levels of Ly49A, Ly49D, and DAP12. PU.1 suppresses the expression of Ets-1 which directly binds the promoter region of *Id2*, *Tbx21*, and *Jak3*. The augmented level of Id2, T-bet, and Jak3 regulate NK cell development in different ways. Ets-1 promotes the expression of NCR1, Ly49D, and Ly49H. The line between PU.1 and Ets1 represents that PU.1 directly binds to Ets-1 promoter region and inhibit Ets-1 transcription. (**B**) During NK cell development, *Nfil3* is repressed by Ets-1, and promotes the transcription of *Id2* and *Eomes*. Nfil3 upregulates NCR1, NKG2D, and DX5 and downregulates Ly49C, Ly49I, Ly49F, and Klrg1. Tox2 directly binds to the promoter region of *Tbx21* and activates its transcription. Tox2 augments the expression of NKR1 to fulfil the NK cell effector function. Runx3 cooperates with Ets-1 to promote NK cell proliferation. Eomes maintains the expression of NK1.1 and NCR1 during NK cell development. Eomes augments the expression of DX5 and Ly49s and represses the expression of TRAIL. Eomes can bind to the *Ifng* promoter region and activate its transcription. T-bet maintains the expression of CD11b, DX5, and Klrg1 but suppresses the expression of c-Kit. T-bet activates transcription of *Gzmb* and *Prf1*. T-bet and Eomes cooperatively bind to the *Il2rb* promoter region. The dashed line between Ets1 and Nfil3 represents that the Ets1 inhibits the transcription of *Nfil3* in an indirect way.

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
