# Peer review of "Transcriptional Regulation of Natural Killer Cell Development and Functions"

_cancers, 2020, doi:10.3390/cancers12061591_

Round 1

Reviewer 1 Report

The present review is well-written and clear, well organized and nicely illustrated.

Even if it is not the main aspect of this review, I feel that there is a major lack of indications of potential therapeutic strategies and when there are some, they are not enough described :

  • In introduction section: “clinical applications include antibody (such as Daratumumab and Elotuzumab) for ADCC”, please indicate the targets of those antibodies, namely CD38 and SLAMF7 respectively.
  • 11.b : “NK cell egress can be rescued following the treatment of CXCR4 antagonists”, please give examples.
  • Many strategies are already developed against Notch, Ikaros, FoxO1,

Other transcription factors were also recently associated with NK cell development such as PBX1 (Xu et al., PBX1 promotes development of natural killer cells by binding directly to the Nfil3 promoter. FASEB J. 2020 May;34(5):6479-6492. doi: 10.1096/fj.202000121R. Epub 2020 Mar 19.), or PLZF (Hess et al., The Transcription Factor Promyelocytic Leukemia Zinc Finger Protein Is Associated With Expression of Liver-Homing Receptors on Human Blood CD56bright Natural Killer Cells. Hepatol Commun. 2020 Jan 6;4(3):409-424. doi: 10.1002/hep4.1463.)

The zinc finger transcription factor Iaolos is better known as IKZF3 for Ikaros family zinc finger protein 3, please indicate this in the review and also give alias names when necessary along the manuscript. However, the same alias should be used as main name (for instance E4BP4 or Nfil3 in Figures 2 and 3).

The different transitions stages and transcription factor functions are well depicted. The only point ids the lack of differences between the different mature stages of NK cells associated with late/terminal stages of NK cells differentiation.

A list or table of abbreviations should be given at the end of the review. Some abbreviations are not included are the right place. For instance “dendritic cells (DCs)” lane 11 of # 3.2. when “DCs” was already written the lane before.

Figure 3: the authors indicate in #4.6. a role of ID2 to convert iNK into more mature NK cells whereas ID2 is indicated in figure 3 as an activator of CLP to NK progenitor development. Is the function of ID2 in iNK to mNK transition limited to mouse model or exist in human? Please comment on this and change accordingly in figure and or #4.6 to precise the differences between mouse or human NK cell development as no species information are yet indicated in #4.6. Please check along the manuscript for such concordances (for instance for RUNX3 among others).

The string network analyses (Figures 4 and 5) might be clearer as hierarchical pictures instead of the presented circle pictures. 

Figure 6 : the “not involved in function gene circle should not cover the 2 other one but be an independent circle for clarity. Please organize the illustration of TF in close proximity of the presented circles, or removes as they are just illustrative and not necessary for the message. The question marks for Tox2 and PU.1 should be discussed in legend.

Figure 7 : why is there a plain inhibition line between PU.1 and Ets1 but a dashed inhibition line between Ets1 and Nfil3 ? Please comment or change.

Some other more recent references should also be added and commented in this review:

  • role of Smad3 transcription factor to inhibit the E4BP4/NFIL3 pathway that control NK cell development (Tang et al, Smad3 promotes cancer progression by inhibiting E4BP4-mediated NK cell development.,Nat Commun. 2017 Mar 6;8:14677. doi: 10.1038/ncomms14677), and potential targeting (Lian et al., Combination of Asiatic Acid and Naringenin Modulates NK Cell Anti-cancer Immunity by Rebalancing Smad3/Smad7 Signaling. Mol Ther. 2018 Sep 5;26(9):2255-2266. doi: 10.1016/j.ymthe.2018.06.016).
  • For FoxO1: Tao et al., Differential Control of iNKT Cell Effector Lineage Differentiation by the Forkhead Box Protein O1 (Foxo1) Transcription Factor. Front Immunol. 2019 Nov 21;10:2710. doi: 10.3389/fimmu.2019.02710.
  • For Id2 function : Gabrielli et al., Murine thymic NK cells are distinct from ILC1s and have unique transcription factor requirements. Eur J Immunol. 2017 May;47(5):800-805. doi: 10.1002/eji.201646871.
  • For the role of TGFb in Eomes and Tbet anagonist activities in NK cell maturation: Harmon et al., Liver-Derived TGF-β Maintains the Eomes hi Tbet lo Phenotype of Liver Resident Natural Killer Cells. Front Immunol. 2019 Jul 3;10:1502. doi: 10.3389/fimmu.2019.01502.
  • Other recent article on Eomes function in NK cell development : Shimizu et al., Eomes transcription factor is required for the development and differentiation of invariant NKT cells. Commun Biol. 2019 Apr 29;2:150. doi: 10.1038/s42003-019-0389-3. eCollection 2019.
  • Other recent reviews on the same topic : Held et al., Transcriptional regulation of murine natural killer cell development, differentiation and maturation, Cellular and Molecular Life Sciences volume 75, pages3371–3379(2018); Kee et al., Transcriptional regulation of natural killer cell development and maturation, Adv Immunol. 2020;146:1-28. doi: 10.1016/bs.ai.2020.01.001.

Author Response

Reviewer #1

The present review is well-written and clear, well organized and nicely illustrated.

Even if it is not the main aspect of this review, I feel that there is a major lack of indications of potential therapeutic strategies and when there are some, they are not enough described :

We appreciate Reviewers comment. We would like to respectfully point out that this manuscript primarily focused on the transcriptional regulation of NK cell development. The therapeutic strategies are beyond the scope of this review and we are happy to contribute separately on this topic.

In introduction section: “clinical applications include antibody (such as Daratumumab and Elotuzumab) for ADCC”, please indicate the targets of those antibodies, namely CD38 and SLAMF7 respectively.

As the reviewer mentioned, Daratumumab is the human anti-CD38 IgG1 monoclonal antibody that specifically binds to CD38 expressed on the surface of multiple myeloma cells [1]. It mediates a broad-spectrum killing activity, including antibody-dependent cellular cytotoxicity (ADCC), complement-dependent cytotoxicity (CDC), antibody-dependent cellular phagocytosis (ADCP), and tumor cell apoptosis [2]. Elotuzumab is the human IgG1 monoclonal antibody targeted to the surface glycoprotein signaling lymphocytic activation molecule F7 (SLAMF7) expressed on multiple myeloma cells [3]. SLAMF7 is also expressed on the surface of NK cells. Elotuzumab elicits anti-tumor actives through a dual mechanism: direct activating NK cells and stimulating robust of ADCC [3]. We have included all this information in the current version of this manuscript.

11.b : “NK cell egress can be rescued following the treatment of CXCR4 antagonists”, please give examples.

In our MS, we mentioned that ‘These Gata3-deficient NK cells are unable to exit from the BM’; and ‘the NK cell egress can be rescued following the treatment of CXCR4 antagonists which recruit iNK from BM cells to blood [4]’. Under this condition, CXCR4 antagonists, AMD3100, can rescue the defect of Gata3-deficient NK cell egression. Previous work showed that the administration of AMD3100 in mice resulted in reduced NK cell numbers in BM and augmented numbers in spleen and peripheral blood [5] . This suggests that the defect of Gata3-deficient NK cell egression is associated with the CXCR4 signaling pathway. Current version of the manuscript includes this information.

Other transcription factors were also recently associated with NK cell development such as PBX1 (Xu et al., PBX1 promotes development of natural killer cells by binding directly to the Nfil3 promoter. FASEB J. 2020 May;34(5):6479-6492. doi: 10.1096/fj.202000121R. Epub 2020 Mar 19.), or PLZF (Hess et al., The Transcription Factor Promyelocytic Leukemia Zinc Finger Protein Is Associated With Expression of Liver-Homing Receptors on Human Blood CD56bright Natural Killer Cells. Hepatol Commun. 2020 Jan 6;4(3):409-424. doi: 10.1002/hep4.1463.)

We appreciate Reviewers comment. We have included the following two paragraphs in the current version of our manuscript.

Pre-B cells leukemia transcription factor 1 (PBX1), a homeodomain transcription factor, was reported playing a role in the NK progenitor stage by only one group recently [6]. Lack of PBX1 in hematopoietic cells results in the decreased number of NK progenitors through directly binding to the Nfil3 promotor region and upregulating its expression [6]. However, there is not enough evidence showing if PBX1 plays a role in the development of mature NK cells or NK cell effector function. We add more information in Nfil3 section as the upstream regulatory molecule in current MS.

Promyelocytic leukemia zinc finger (PLZF, encoded by Zbtb16 gene), a member of BTB-zinc finger TF family, plays important roles in the lineage commitment, development, and effector functions of lymphocytes, including CD56bright NK cells [7], innate lymphoid cell 1 (ILC1) [8], NKT [9,10], and gd T cell [11]. PLZF is able to be detected on both human and mice NK cells [7,12]. Human adaptive NK cells display significantly downregulated expression of PLZF, which suggests that lack of PLZF marks gain of human adaptive NK cells [13]. PLZF-/- mice display the defect of IFN-g production but act as a dispensable TF for NK cell development [12]. In addition, PLZF is highly associated with markers of human liver residency NK cells [7].

The zinc finger transcription factor Iaolos is better known as IKZF3 for Ikaros family zinc finger protein 3, please indicate this in the review and also give alias names when necessary along the manuscript. However, the same alias should be used as main name (for instance E4BP4 or Nfil3 in Figures 2 and 3).

As reviewer indicated, Aiolos, as a transcription factor, is also known as Ikaros family zinc finger protein 3 (IKZF3). We indicate this in the current version of our manuscript.

For the name of E4BP4 and Nfil3, we changed to the same name in figures.

The different transitions stages and transcription factor functions are well depicted. The only point is the lack of differences between the different mature stages of NK cells associated with late/terminal stages of NK cells differentiation.

We agree with the Reviewer’s comment. Our knowledge related to distinct transcription factors that operate at the late/terminal stages are minimal. Role of T-bet and Gata2 were established more than 15 years ago; however, the functions of multiple other transcription factors are still lacking.

A list or table of abbreviations should be given at the end of the review. Some abbreviations are not included are the right place. For instance, “dendritic cells (DCs)” lane 11 of # 3.2. when “DCs” was already written the lane before.

We apologize for the omission. We have fixed this issue in the current version of the manuscript and have also included a list of abbreviations used in the text at the end of the manuscript.

Figure 3: the authors indicate in #4.6. a role of ID2 to convert iNK into more mature NK cells whereas ID2 is indicated in figure 3 as an activator of CLP to NK progenitor development. Is the function of ID2 in iNK to mNK transition limited to mouse model or exist in human? Please comment on this and change accordingly in figure and or #4.6 to precise the differences between mouse or human NK cell development as no species information are yet indicated in #4.6. Please check along the manuscript for such concordances (for instance for RUNX3 among others).

We apologize for the confusion. We have fixed this issue both in the text and in the Figure 3. The high level of ID2 promotes human NK cell development and result in the increased of NKPs. Lack of Id2 in mice causes a blockade at the post iNK and pre-mNK cell stages in mice BM. These suggest that Id2 plays a role in a different way at early NK cells development in the BM but is not an essential TF in NKP stage. As reviewer suggested, we change it in both figures and current MS.  

The string network analyses (Figures 4 and 5) might be clearer as hierarchical pictures instead of the presented circle pictures. 

We appreciate the suggestion. However, at present, we or the String Database possess the ability to present these networks in a linear fashion. As we showed in Figure 4 and 5, the relationship of molecules from networks where TFs are involved are complicated. It is also true that not all the indicated factors operate in a linear manner. The purpose these two figures is to provide an unbiased list of transcription factor interactions.

Figure 6 : the “not involved in function gene circle should not cover the 2 other one but be an independent circle for clarity. Please organize the illustration of TF in close proximity of the presented circles or removes as they are just illustrative and not necessary for the message. The question marks for Tox2 and PU.1 should be discussed in legend.

We have reorganized Figure 6 as reviewer suggested. The question mark in Figure 6 stand for no direct evidence showing the effect on IFN-g production. While, we can infer that it may affect IFN-g production based on current evidence. We discuss this point in Figure 6 legend.

Figure 7 : why is there a plain inhibition line between PU.1 and Ets1 but a dashed inhibition line between Ets1 and Nfil3? Please comment or change.

In Figure 7, the plain inhibition line between PU.1 and Ets1 means that PU.1 directly binds to Ets1 promoter region and inhibit Ets1 transcription. The dashed inhibition line between Ets1 and Nfil3 stands for the Ets1 inhibits the transcription of Nfil3 in an indirect way. We have included these descriptions in the figure legends of the current version of the manuscript.

Some other more recent references should also be added and commented in this review:

  • role of Smad3 transcription factor to inhibit the E4BP4/NFIL3 pathway that control NK cell development (Tang et al, Smad3 promotes cancer progression by inhibiting E4BP4-mediated NK cell development.,Nat Commun. 2017 Mar 6;8:14677. doi: 10.1038/ncomms14677), and potential targeting (Lian et al., Combination of Asiatic Acid and Naringenin Modulates NK Cell Anti-cancer Immunity by Rebalancing Smad3/Smad7 Signaling. Mol Ther. 2018 Sep 5;26(9):2255-2266. doi: 10.1016/j.ymthe.2018.06.016).
  • For FoxO1: Tao et al., Differential Control of iNKT Cell Effector Lineage Differentiation by the Forkhead Box Protein O1 (Foxo1) Transcription Factor. Front Immunol. 2019 Nov 21;10:2710. doi: 10.3389/fimmu.2019.02710.
  • For Id2 function : Gabrielli et al., Murine thymic NK cells are distinct from ILC1s and have unique transcription factor requirements. Eur J Immunol. 2017 May;47(5):800-805. doi: 10.1002/eji.201646871.
  • For the role of TGFb in Eomes and Tbet anagonist activities in NK cell maturation: Harmon et al., Liver-Derived TGF-β Maintains the Eomes hi Tbet lo Phenotype of Liver Resident Natural Killer Cells. Front Immunol. 2019 Jul 3;10:1502. doi: 10.3389/fimmu.2019.01502.
  • Other recent article on Eomes function in NK cell development : Shimizu et al., Eomes transcription factor is required for the development and differentiation of invariant NKT cells. Commun Biol. 2019 Apr 29;2:150. doi: 10.1038/s42003-019-0389-3. eCollection 2019.
  • Other recent reviews on the same topic : Held et al., Transcriptional regulation of murine natural killer cell development, differentiation and maturation, Cellular and Molecular Life Sciences volume 75, pages3371–3379(2018); Kee et al., Transcriptional regulation of natural killer cell development and maturation, Adv Immunol. 2020;146:1-28. doi: 10.1016/bs.ai.2020.01.001.

We thank the reviewer and we have added all these references in the current version of the manuscript.

Reviewer 2 Report

Overall the authors have a very detailed description of the literature of the transcription factors that involved in NK cell development and differentiation in regards to cytokines. This description is intricate and comprehensive. That said there are some big picture things that the authors did not include in the review.

Major:

The authors focus on the role of cytokines and subsequent transcription factors that shape the NK development. Three things missing from the review are 1. how NK eduction/licensing affect transcription factors 2. the role of transcription factors in adaptive NK cells (PLZF). 3. discussion on how the transcription factors can differentiate ILC1s from conventional NK cells. These topics are more recent in the NK field and I think would be of interest to the field. If the authors could add one major section or three subsections dealing with these non-cytokine mechanisms of development as it relates to transcription factors, that would make the review better

Minor:

The authors state "NK cells primarily mediate their functions through four distinct effector mechanisms. They are 1) releasing cytolytic molecules such as perforin and granzymes; 2) mediating antibody-dependent cell cytotoxicity (ADCC); 3) engaging in apoptotic pathways; 4) producing pro-inflammatory cytokines." This is a jumbling of effector mechanisms and recognition systems NK use. I think this should be restated that NK cells recognize cells for killing via 4 mechanisms; natural cytotoxicity, ADCC, TRAIL, and Fasl. NK cells effector functions are mediated by release of lytic granules containing perforin and granzymes that cause apoptosis, binding TRAIL or Fasl on target cells that also causes apoptosis, and both inflammatory and inhibitory cytokine secretion (IFNg/TNFa (go further if want) and IL-10 respectively). 

Changes in these areas would make the manuscript favorable to publish. 

Author Response

Comments and Suggestions for Authors

Overall the authors have a very detailed description of the literature of the transcription factors that involved in NK cell development and differentiation in regards to cytokines. This description is intricate and comprehensive. That said there are some big picture things that the authors did not include in the review.

Major:

The authors focus on the role of cytokines and subsequent transcription factors that shape the NK development. Three things missing from the review are 1. how NK education/licensing affect transcription factors 2. the role of transcription factors in adaptive NK cells (PLZF). 3. discussion on how the transcription factors can differentiate ILC1s from conventional NK cells. These topics are more recent in the NK field and I think would be of interest to the field. If the authors could add one major section or three subsections dealing with these non-cytokine mechanisms of development as it relates to transcription factors, that would make the review better.

NK cell licensing is defined as a process that NK cells undergo a functional maturation process depend on major histocompatibility (MHC) class I [14]. In homeostasis, NK cell inhibitory receptors recognize autologous MHC class I molecules, where NK cells will contain their activation in effector responses. Thus, licensed NK cell effector function is inhibited by MHC class I.  Under inflammatory conditions, the inhibitory receptors on the target cells are missing and not able to be recognized. NK cells are unlicensed NK cells and are activated to cause effector functions.

Promyelocytic leukemia zinc finger (PLZF, encoded by Zbtb16 gene) is a member of BTB-zinc finger TF family. PLZF is able to be detected on both human and mice NK cells [7,12]. Human adaptive NK cells display significantly downregulated expression of PLZF, which suggests that lack of PLZF marks gain of human adaptive NK cells [13]. PLZF-/- mice display the defect of IFN-g production but act as a dispensable TF for NK cell development [12]. In addition, PLZF is highly associated with markers of human liver residency NK cells [7].

Zbtb32 is another TF from BTB-zinc finger TF family. The expression of Zbtb32 is induced by MCMV infection, which is required for NK cell expansion after infection [15]. In Zbtb32-deficient mice, the percentage and number of mature NK cells is comparable with WT mice[15]. And Zbtb32-/- mice display normal expression profile of NK cell identity and function markers, including Ly49H, Ly49D, NK1.1, DX5, Ly49C, Ly49I, and Ly49A [15]. Both WT and Zbtb32-/- mice shows similar upregulation of cytotoxic protein granzyme B after murine cytomegalovirus (MCMV) infection [15]. Zbtb32 regulates the NK cell antiviral effector function through directly binding and upregulating IRF8, which is a TF required for NK cell antiviral function and virus-driven expansion [16].

Group 1 innate lymphoid cells (ILCs) is comprised of NK cells and ILC1s, which share several common features: producing IFN-g and requiring T-bet for their functions . However, ILC1s are non-cytotoxic, tissue-resident, and non-Eomes dependent ILCs [17-19].  Common innate lymphoid progenitors (CILPs) differentiated from CLPs, develop into either NKPs or Innate lymphoid cell precursors (ILCP) [19,20].  The transcription factors of Gata3 and PLZF play critical roles in the differentiation of ILC1s from conventional NK cells [8,21]. Deletion of PLZF significantly altered the development of ILCs subset, but not NK cell development [20]. Conditional Gata3 knockout in hematopoietic cells (Gata3fl/fl-VavCre/+) showed that significantly reduced PLZF+ ILCPs [22].

These three paragraphs are added to the current manuscript.

Minor:

The authors state "NK cells primarily mediate their functions through four distinct effector mechanisms. They are 1) releasing cytolytic molecules such as perforin and granzymes; 2) mediating antibody-dependent cell cytotoxicity (ADCC); 3) engaging in apoptotic pathways; 4) producing pro-inflammatory cytokines." This is a jumbling of effector mechanisms and recognition systems NK use. I think this should be restated that NK cells recognize cells for killing via 4 mechanisms; natural cytotoxicity, ADCC, TRAIL, and Fasl. NK cells effector functions are mediated by release of lytic granules containing perforin and granzymes that cause apoptosis, binding TRAIL or Fasl on target cells that also causes apoptosis, and both inflammatory and inhibitory cytokine secretion (IFNg/TNFa (go further if want) and IL-10 respectively). 

We truly appreciate Reviewer’s suggestion. Accordingly, we have changed the way we stated these.

Changes in these areas would make the manuscript favorable to publish. 

We have made the required changes. Thank you.

Reviewer 3 Report

The manuscript is well written, easy to understand and reviews the recent advances in the knowledge of the trascriptional factors involved in generation and function of NK cells.

I would suggest to create a table summarizing the potential role in tumors of the several TFs (at least for those already know) in order to recapitulate the complexity of the network.

Author Response

The manuscript is well written, easy to understand and reviews the recent advances in the knowledge of the trascriptional factors involved in generation and function of NK cells.

I would suggest to create a table summarizing the potential role in tumors of the several TFs (at least for those already know) in order to recapitulate the complexity of the network.

We apologize to the Reviewer that we are unable to add this table. This review is focused on the role of TFs in NK cell development. As reviewer mentioned, a summary of the potential role of TFs in NK cell effector functions has been displayed in Figure 6, where we summarize the specific effector functions of each TF. We will include such a table including the transcription factors that directly regulate the functions of NK cells in our future work.

Round 2

Reviewer 2 Report

The authors more than adequately addressed my concerns. I support this nice review to be published.